# Does Fertilizer Education Program Increase the Technical Efficiency of Chemical Fertilizer Use? Evidence from Wheat Production in China

**Pingping Wang [1]** , **Wendong Zhang [2]** , **Minghao Li [2]** and **Yijun Han [1],***

[1]  College of Economics and Management, China Agricultural University, Beijing 100083, China; wppprivate@126.com

[2]  Department of Economics and Center for Agricultural and Rural Development, Iowa State University, Ames 50010, USA; wdzhang@iastate.edu (W.Z.); minghao@iastate.edu (M.L.)

*  Correspondence: hyjcau@126.com; Tel.: +86-010-6273-8685

**Abstract:** Farmers in China and many other developing countries suffer from low technical efficiency of chemical fertilizer use, which leads to excessive nutrient runoff and other environmental problems. A major cause of the low efficiency is lack of science-based information and recommendations for nutrient application. In response, the Chinese government launched an ambitious nationwide program called the "Soil Testing and Fertilizer Recommendation Project" (STFRP) in 2005 to increase the efficiency of chemical fertilizer use. However, there has been no systematic evaluation of this program. Using data from a nationally representative household survey, and using wheat as an example, this paper first quantifies the technical efficiency of chemical fertilizer use (TEFU) by conducting stochastic frontier analysis (SFA), then evaluates the impact of STFRP on the TEFU using a generalized difference-in-difference approach. We found that STFRP, on average, increased TEFU in wheat production by about 4%, which was robust across various robustness checks. The lessons learned from STFRP will be valuable for China's future outreach efforts, as well as for other countries considering similar nutrient management policies.

**Keywords:** technical efficiency of chemical fertilizer use; nutrient management; difference-in-difference (DID); stochastic frontier analysis (SFA); agricultural input efficiency; China

---

## 1. Introduction

The efficiency of chemical fertilizer use is very low in China. The average chemical fertilizer usage per hectare in China increased from 86.72 kg/ha in 1980 to 359.08 kg/ha in 2016, about 3.3 times that of the United States and 3.6 times that of the world average [1]. The chemical fertilizer utilization rate (the ratio of the amount absorbed by crops to the total amount applied) in grain production is only 35.2% [2], and the technical efficiency of chemical fertilizer use (TEFU hereafter) is only around 0.3 [3–5]; this suggests that the actual fertilizer application rate is more than three times the optimal fertilizer usage level. The low efficiency of chemical fertilizer use results in eutrophication in streams and rivers [6], groundwater nitrate pollution [7], high greenhouse gas emissions [8], and loss of biodiversity [9]. For example, due to the low efficiency of chemical fertilizer use, more than 92% of lakes in China have been affected by chemical fertilizer runoff [10].

Wheat, which is one of the most important food crops in China, has an even more severe chemical fertilizer use efficiency problem than other grain crops. Wheat accounts for 27% of China's crop planting area, with the intensity of chemical fertilizer reaching 405 kg/ha, compared to 345 kg/ha for rice and 375 kg/ha for corn [11].

Many factors lead to the low efficiency of Chinese farmers' chemical fertilizer use, including risk aversion, lack of on-farm labor [12], and low fertilizer prices caused by heavy government fertilizer subsidies [13,14]. Of most relevance to this study, previous research has often cited farmers' lack of knowledge regarding chemical fertilizers and their use as a primary cause [15]. Many Chinese farmers assume that more fertilizers lead to higher crop yields, without realizing that a plateau has been reached.

To address the problem of low chemical fertilizer use efficiency, the Chinese government launched an ambitious nationwide program called the "Soil Testing and Fertilizer Recommendation Project (STFRP)" in 2005, which covered wheat and other major crops. As the name suggests, the STFRP mobilized thousands of governmental extension personnel to systematically conduct soil tests and field trials across the nation to determine county-specific fertilizer usage levels and timing for major row crops and different types of fertilizers, including nitrogen (N), phosphorus (P), and potassium (K). Soil test results were then formulated into nutrient application recommendations that were disseminated among farmers via training courses, demonstrations, street signs, and coverage by local newspaper and television outlets [16]. For example, in Gongyi County in Henan Province, the targeted wheat yield was 4260 kg/ha. Fertilizers were recommended to be applied twice during the growth process, with a maximum of 203 kg/ha of the fertilizer (including 26% N, 28% P and 10% K) before sowing and 69 kg/ha of the nitrogen fertilizer during the jointing stage (the jointing stage is a critical phase of wheat development and is marked by the first joint's or node's emergence above the soil line), which is 33% lower than the current prevailing application rate.

In 2005, the Chinese government chose 200 major agricultural counties in 13 major grain-producing provinces in which the STFRP was implemented. In 2006, another 400 major agricultural counties were chosen, with previously enrolled counties continuing to implement the program. By 2009, the number of counties enrolled in STFRP had reached 2498, with more than 90% of major wheat-producing counties enrolled. STFRP is the most important project implemented by the Chinese government to increase chemical fertilizer efficiency; yet, despite its importance, there has been no rigorous evaluation of the impact of STFRP on the TEFU for wheat.

While the agronomic significance of optimal fertilizer application is well established [17], the real-world impacts of research-based fertilizer application recommendations are less clear. For fertilizer research to actually change fertilizer usage, research results must be translated into recommendations, the recommendations must be communicated to farmers through educational programs, and farmers must follow the recommendations. It is of great importance to evaluate the overall impacts of fertilizer education to understand how scientific research induces actual changes in policy-making and the subsequent behavior of individual farmers. Specifically, rigorous evaluation of STFRP is necessary because the program is a major fertilizer education effort, with the potential to change fertilizer application practices nationwide. Furthermore, as a case study, the results of STFRP will be a valuable lesson for other farmer education programs in China, such as the 2017 "manure instead of chemical fertilizer program", which is a new manure testing and recommendation program aimed at promoting the use of organic fertilizer as an alternative to chemical fertilizers. So far, the program has been implemented in 200 counties in China for fruit, vegetables, and tea. Study results will also be valuable for other developing countries, especially in Africa, where for instance information and communication technology (ICT) projects have shown that radio broadcasting was an effective way to communicate agricultural knowledge and change farmers' behavior, such as improving soil by using composted manure and other conservation practices [18]). The importance of program evaluations, such as Qiu and He's study [19], is widely recognized by the scientific community as an important component of scientific research.

Exploiting the gradual enrollment of households into STFRP and a large nationally representative rural household survey, this study aims to evaluate the impacts of STFRP on the TEFU for wheat production measured by stochastic frontier analysis (SFA). Using a generalized difference-in-difference

(DID) specification [20], we found that on average, STFRP significantly increased TEFU in wheat production by about 4%, and this result was robust to various robustness checks.

## 2. Materials and Methods

### 2.1. Using SFA to Measure TEFU

The purpose of this paper is to measure the impacts of STFRP on the TEFU of wheat production in China. In production economics, technical efficiency is defined as the effectiveness with which a given set of inputs is used to produce an output [21]. For agricultural production, given a certain quantity of outputs, technical efficiency is achieved when we use the minimum input possible. Specifically, the TEFU measures the ratio of the minimum chemical fertilizer used to the actual chemical fertilizer used when producing a certain quantity of outputs. This definition is related to, but different from, the agronomical definition of fertilizer use efficiency, which is the ratio of chemical fertilizer absorbed by the plant to actual usage [22]. The TEFU has a solid theoretical foundation in economic theory [3] and has been routinely estimated in empirical studies, usually using producer (in our case, rural household) surveys [3,5,23].

We used SFA, a well-established method for estimating technical efficiency, to measure TEFU. Pioneering work [24] provides the definition of and conceptual framework for technical efficiency. In the past several decades, many researchers [25,26] have led the effort to further develop SFA, and it has been widely applied in the empirical literature [3,15].

In this paper, we followed Battese and Coelli's research [25] and considered three inputs in wheat production—labor (L), chemical fertilizer (F), and other inputs (O)—and one output, wheat yield (Y). Because we used the input per hectare or output per hectare to express the production function, the land input did not explicitly appear in the model. The general SFA model can be expressed as:

$$nY_{it} = lnf(L_{it}, F_{it}, O_{it}; \beta) + \gamma_{it} - \mu_{it} \tag{1}$$

where subscript $i$ indicates the $i^{th}$ farmer; $t$ represents year; $\beta$ is the vector of parameters to be estimated; $\gamma_{it}$ is the random error, which is assumed to be distributed normally with mean 0 and variance $\sigma_\gamma^2$; and $\mu_{it}$ captures inefficiency during production and is assumed to be nonnegative. It is assumed that $\mu_{it}$ and $\gamma_{it}$ are independent. Two specifications of $\mu_{it}$ are commonly used in SFA models: one is for $\mu_{it}$ to follow a time-invariant truncated normal random distribution with mean $\mu$ and variance $\sigma_u^2$; the other is the time-varying decay specification, where $\mu_{it} = \exp\{-\eta(t - T_i)\}\mu_i$, with $T_i$ being the last period for the $i^{th}$ panel (farmer); $\eta$ being the decay parameter; and $\mu_i$ drawn from normal distribution with mean $\mu$ and variance $\sigma_u^2$. In this paper, we used the time-varying specification in the main analysis and test for the time-invariant specification (Table 2).

The TEFU for farmer $i$ at time $t$ was defined as the ratio of the minimum amount of chemical fertilizer required ($F^*$) divided by the observed chemical fertilizer input ($F$), other things being equal [3,20]. Specifically:

$$TEFU_{it} = \min\{\theta : f(L_{it}, F^*_{it}, O_{it}) = f(L_{it}, \theta F_{it}, O_{it}) \geq Y_{it}\} \leq 1 \tag{2}$$

where $\theta$ represents the technical efficiency of fertilizer usage ($TEFU_{it}$) and $\theta = 1$ indicates that chemical fertilizer use is efficient and that the chemical fertilizer input reaches a technically efficient frontier while holding all other inputs at observed levels.

The property of duality in production economics determines that if farmer $i$'s fertilizer use is efficient, the production process is also technically efficient with $\mu_{it}$ equal to zero [3]. Thus, Equation (1) can be written as:

$$lnY_{it} = lnf(L_{it}, F^*_{it}, O_{it}; \beta) + \gamma_{it} \tag{3}$$

Combining Equations (1) and (3), we can obtain Equation (4):

$$lnf(L_{it}, \theta F_{it}, O_{it}; \beta) - lnf(L_{it}, F_{it}, O_{it}; \beta) + \mu_{it} = 0 \tag{4}$$

Equation (4) can be used to estimate $\theta$, the measure of TEFU, where $\theta$ equal to 1 means that chemical fertilizer use is technically efficient, i.e., the observed chemical fertilizer input is equal to the minimum amount of chemical fertilizer required when other things are equal. For example, $\theta = 0.5$ means that $TEFU_{it}$ is 0.5 and the observed chemical fertilizer input is two times the minimum amount of chemical fertilizer required when other things are equal. A common estimation technique is to specify the following translog production function [26,27]. In addition, the log-likelihood ratio tests that compare translog production functions with alternative specifications, shown in Table 2, suggest that the translog production function with time-varying $\mu_{it}$ is the most preferred specification for our data. Additional details about these specification tests are provided at the results section later.

$$
\begin{aligned}
lnY_{it} = \beta_0 + \beta_1 lnF_{it} \quad & + \beta_2 lnL_{it} + \beta_3 lnO_{it} + \beta_4 t + \beta_5 (lnF_{it})^2 + \beta_6 (lnL_{it})^2 + \beta_7 (lnO_{it})^2 \\
& + \beta_8 t^2 + \beta_9 lnF_{it} * lnL_{it} + \beta_{10} lnF_{it} * lnO_{it} + \beta_{11} lnO_{it} * lnL_{it} + \beta_{12} t \\
& * lnF_{it} + \beta_{13} t * lnL_{it} + \beta_{14} t * lnO_{it} + \gamma_{it} - \mu_{it}
\end{aligned} \tag{5}
$$

Combining Equations (4) and (5) yields the following relationship:

$$a\,(ln\theta)^2 + b\,ln\theta + c = 0 \tag{6}$$

where $a = \beta_5$, $b = \beta_1 + 2\beta_5\,lnF_{it} + \beta_9\,lnL_{it} + \beta_{10}\,lnO_{it} + \beta_{12}\,t$, and $c = \mu_{it}$.

The variable $ln\theta$ in Equation (6) measures TEFU and can be obtained using the quadratic root formula in Equation (7):

$$ln(TEFU_{it}) = ln\theta = \left(-b + \sqrt{b^2 - 4ac}\right)/2a \tag{7}$$

We finally estimated Equation (5) using household panel data to get a, b, and c, and then calculated $TEFU_{it}$ using Equation (7).

*2.2. DID Analysis*

We used the DID method to assess the effect of STFRP on TEFU. The DID method is well established in economics for evaluating program impacts [28–30]. It estimates the treatment effect by comparing changes in the treated group (STFRP households) to changes in the control group (non-STFRP households) while controlling for unobserved variables that are fixed over time (e.g., geographical location) and common to all households at a given time (e.g., regional wheat price). Given that the STFRP was gradually rolled out, we had multiple treatments applied at different times. The standard specification for this setup is the generalized DID model [20], which is also known as staggered DID [31,32]).

In our context, the generalized DID model was specified as follows:

$$TEFU_{it} = \text{Constant} + \delta T_{it} + \pi ln(X_{it}) + \alpha_i + \lambda_t + \epsilon_{it} \tag{8}$$

where subscript *i* indicates the $i^{th}$ farmer and *t* represents year; $TEFU_{it}$ is the technical efficiency of fertilizer use; The treatment variable $T_{it}$ equals one if household *i* has enrolled in STFRP in year *t* and zero otherwise; The coefficient $\delta$ of the treatment variable $T_{it}$ measures the effect of STFRP on the TEFU through comparisons between treated households and control households; $X_{it}$ represents control variables for household or field characteristics, $\alpha_i$ is the household fixed effects, $\lambda_t$ is year fixed effect; and $\epsilon_{it}$ is an error term. We used OLS (Ordinary Least Square) with robust errors clustered at the county level. Because $TEFU_{it}$ is truncated from below at 0 and above at 1, we also use Tobit as a robustness check.

We carried out two placebo tests to validate our results. First, we artificially moved STFRP implementation to two years before the year of actual implementation. The TEFU should only increase from the year when STFRP was first implemented, not from two years earlier. Second, we examined the impact of STFRP on pesticide use. Since STFRP targets chemical fertilizer use rather than pesticide use, this should not lead to changes in pesticide application rates.

## 2.3. Data and Descriptive Statistics

The National Fixed Point Survey (NFPS) dataset is jointly collected by the Chinese Research Center of Rural Economy within the Ministry of Agriculture and the Rural Affairs and Chinese Central Policy Research Office. It has been a nationally representative annual household survey of 20,398 rural households in 335 villages across 28 provinces in China since 1986. The survey is the most comprehensive dataset on Chinese agriculture, and Chinese policy-makers rely on it to gauge agricultural production and rural development. To our knowledge, this study is the first to use these data to evaluate the impact of STFRP.

In this paper, we used a subsample from NFPS that contains 2054 households from four major wheat-producing provinces in China (Shanxi, Henan, Hubei, and Sichuan). Our study area covers almost half of China's major wheat-producing provinces, which account for more than one-third of the nation's wheat production. Our dataset spans from 2003 to 2008. We dropped 917 observations with missing data on key variables, such as the wheat planting area, wheat fertilizer expenditure, education level, and total income. Data on fertilizer expenditure, seed cost, pesticide cost, and irrigation cost are trimmed at the 1st and 99th percentiles of the distributions to remove outliers.

The timing of STFRP implementation was collected from local government documents. The STFRP was launched in 2005 and has been gradually implemented over the years. We went through hundreds of government documents online to identify the timing of STFRP implementation for each county in our data; more details are available in Appendix A Table A1. It should be noted that the STFRP was first implemented 2005, and by 2009 more than 90% of the major wheat-producing counties had implemented STFRP. Since the main feature of the program is to provide information, we expect that most of the impacts happen at initial implementation, which will be captured by our evaluation of the early implementation years. Once farmers received the initial recommendation, later recommendations would only fine-tune the initial recommendation and their effects will be smaller.

Table 1 presents summary statistics for key variables of our paper. While most of the statistics are self-explanatory, two items must be highlighted. First, average annual expenditure on fertilizers is 131 US dollars per hectare, which accounts for 35% of the total variable production cost (total variable production cost includes other expenditure and fertilizer expenditure, but not the implicit cost of farmers' labor). Using the chemical fertilizer price for each province [33], we also converted chemical fertilizer expenditure to chemical fertilizer usage. From 2003 to 2008, Chinese wheat farmers used about 305 kg of chemical fertilizer per hectare annually. All prices are normalized to the year 2003 using regional price indices from the National Bureau of Statistics [11].

**Table 1.** Summary statistics (2003–2008, number of Households = 1137).

| Variable | Description | Mean | S.D. | Min | Max |
|---|---|---|---|---|---|
| Y | Wheat yield (kg/ha) | 5775.89 | 2209.91 | 675.00 | 13,500.00 |
| L | Labor input (day/ha) | 345.88 | 268.43 | 19.10 | 1875.00 |
| F | Fertilizer expenditure (dollar/ha) | 130.48 | 71.05 | 13.86 | 402.67 |
| F1 | Fertilizer usage (kg/ha) | 304.48 | 165.91 | 32.87 | 938.34 |
| O | Other expenditure (dollar/ha) | 240.00 | 116.26 | 2.14 | 932.68 |
| Age | Age of the head of household (year) | 51.67 | 10.80 | 28.00 | 79.00 |
| Edu | Years of education (years) | 6.61 | 2.53 | 0.00 | 12.00 |
| Fasset | Fixed assets for agricultural production (dollar) | 631.36 | 2019.61 | 0.00 | 80,259.13 |

**Table 1.** *Cont.*

| Variable | Description | Mean | S.D. | Min | Max |
|---|---|---|---|---|---|
| Tland | Household's total land area (hectare) | 0.35 | 0.35 | 0.00 | 2.67 |
| Tincome | Household's total income (dollar) | 2739.90 | 2692.35 | 244.29 | 35,033.41 |
| Wharea | Wheat planting area (hectare) | 0.16 | 0.18 | 0.00 | 4.67 |

Note: Other expenditure includes seed, pesticide, manure, and machine and irrigation cost.

## 3. Results

### 3.1. TEFU Estimation

Table 2 presents the results of several alternative specifications of the SFA model compared to the preferred specification: translog production function with time-varying inefficiency ($\mu_{it}$) (Equation (5)). These alternative specifications included the Cobb–Douglas production function, translog production function with no technical change, translog production function with neutral technical change, and translog production function with time-invariant $\mu_{it}$. Log-likelihood ratio tests rejected the null hypotheses for the alternative models, suggesting that the translog production function with time-varying $\mu_{it}$ was required for our data.

**Table 2.** Model specification tests for SFA analysis.

| Model | Null Hypothesis | Log-likelihood | LR Test | $\chi^2_{0.05}$ |
|---|---|---|---|---|
| Cobb–Douglas production function | $\beta_5 \cdots \beta_{14} = 0$ | −1148.42 | 135.16 | 19.68 |
| Translog production function with no technical change | $\beta_4, \beta_8, \beta_{12} \cdots \beta_{14} = 0$ | −1158.56 | 155.43 | 12.59 |
| Translog production function with neutral technical change | $\beta_{12} \cdots \beta_{14} = 0$ | −1113.83 | 65.96 | 9.49 |
| Translog production function with time-invariant inefficiency | $\eta = 0$ | −1094.96 | 28.23 | 3.84 |

Note: Critical values ($\chi^2_{0.05}$) for the tests were obtained from Table 1 of Kodde and Palm's paper [34], where the degrees of freedom were q+1 and q was the number of parameters specified to be zero.

Parameter estimates of our preferred SFA model are reported in Table 3. Most estimated coefficients were statistically significant. $\eta$ was significant at the 1% level, meaning that our data required a time-varying decay model ($\mu_{it}$ was time-varying). These results provided further support for our specification of the production function.

Figure 1 shows the annual kernel density distribution of the estimated TEFU in STFRP counties and non-STFRP counties from 2005 to 2008. This served as a descriptive comparison of TEFU for counties enrolled in the program versus those that were not, and revealed that TEFU was skewed to the lower end of the spectrum. Specifically, the mean of TEFU for all areas was about 0.17, which meant that on average the actual fertilizer rate was 4.9 times higher than the minimum usage, suggesting that fertilizer usage was far from the technical production frontier. The TEFU we estimated seemed low. This was possibly because we were comparing across locations without taking soil quality and climate into account. Some farmers appeared inefficient in the analysis not because they used too much fertilizer, but because the location was not productive. However, this result was consistent with previous studies [3,5,35]. Based on the comparison of Figure 1a–d, the distribution of TEFU for the STFRP area moved to the right over time, while the distribution of TEFU for the non-STFRP area remained skewed toward the left. These figures offered suggestive evidence that STRFR was effective in increasing TEFU. In the following section, we present a more rigorous assessment using the DID method.

**Table 3.** Parameter estimates of the SFA model.

| Variables | Coefficients | Standard Errors | Variables | Coefficients | Standard Errors |
|---|---|---|---|---|---|
| Intercept | 3.599 *** | 0.380 | $lnLlnF$ | −0.056 *** | 0.011 |
| $lnL$ | 0.367 *** | 0.081 | $lnLlnO$ | −0.004 | 0.016 |
| $lnF$ | 0.482 *** | 0.098 | $lnOlnF$ | −0.071 *** | 0.021 |
| $lnO$ | 0.079 | 0.133 | $tlnL$ | −0.025 *** | 0.003 |
| $t$ | 0.177 *** | 0.030 | $tlnF$ | 0.010 ** | 0.004 |
| $(lnL)^2$ | −0.008 | 0.006 | $tlnO$ | −0.019 *** | 0.006 |
| $(lnF)^2$ | 0.012 | 0.010 | η | −0.040 *** | 0.008 |
| $(lnO)^2$ | 0.044 *** | 0.016 | Log likelihood | −1,080.844 | |
| $t^2$ | −0.002 | 0.001 | | | |

Note: *** $p < 0.01$, ** $p < 0.05$.

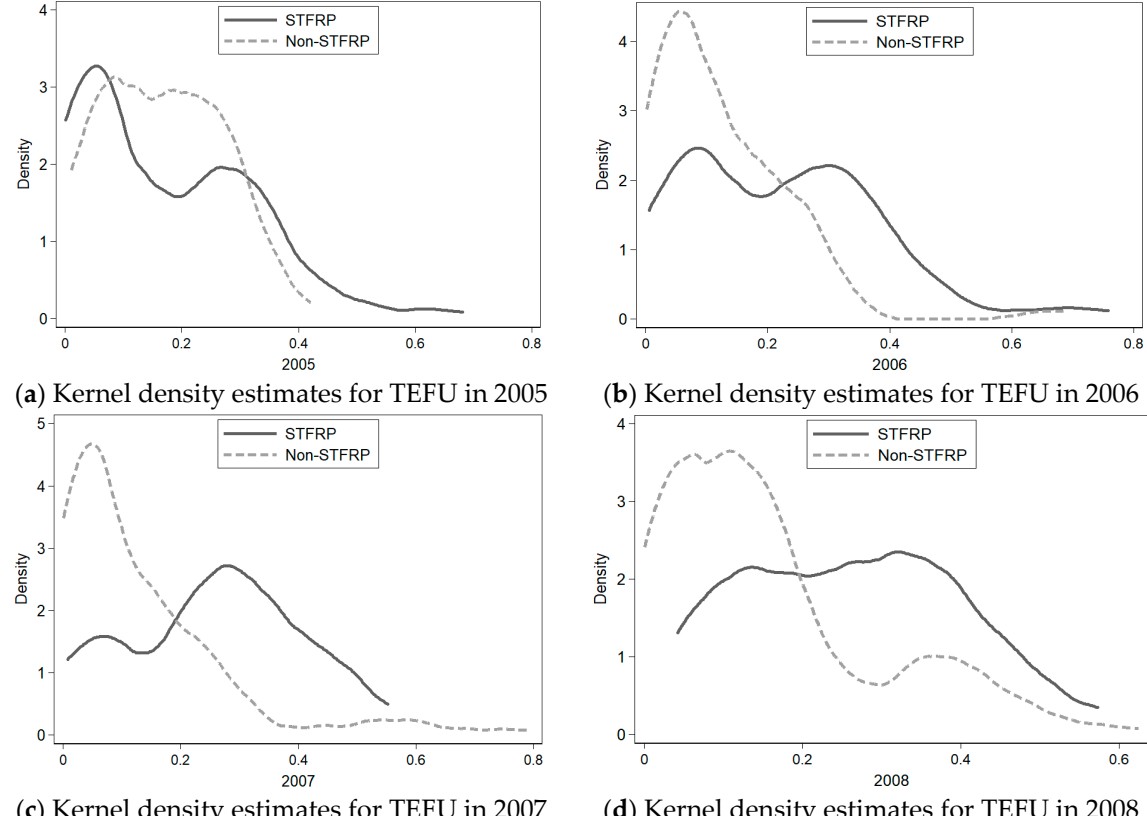

(**a**) Kernel density estimates for TEFU in 2005　　(**b**) Kernel density estimates for TEFU in 2006

(**c**) Kernel density estimates for TEFU in 2007　　(**d**) Kernel density estimates for TEFU in 2008

**Figure 1.** Kernel density estimates for TEFU, 2005-2008. Note: the x-axis is the TEFU.

Figure 2 shows major wheat-producing provinces and average county-level TEFU in the year 2007 as an example. In 2007, the average TEFU in most counties was less than 0.2, with a few exceptions in Henan and Hubei Provinces.

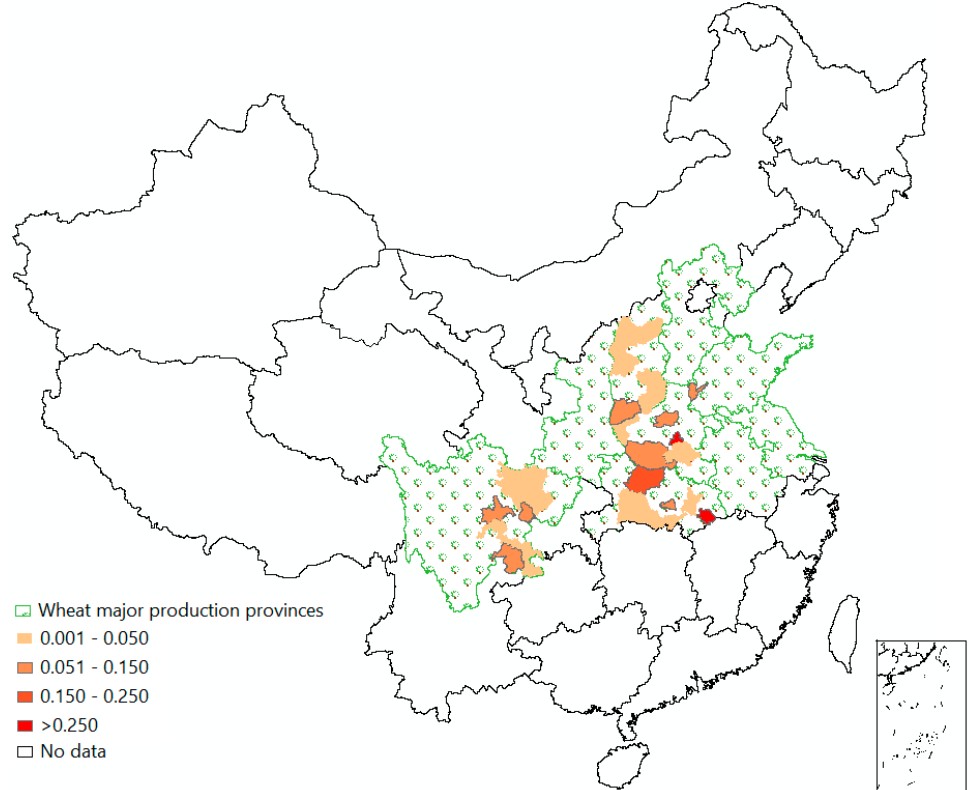

**Figure 2.** TEFU in the year 2007.

### 3.2. DID Analysis of the Effects of STFRP

Table 4 presents the average treatment effect of STFRP on the TEFU of Chinese wheat production from 2005 to 2008 using a general DID approach. All models shown in Table 4 used the TEFU as the dependent variable and included the logarithmic form of household characteristics and field characteristics, such as age, education, seed cost, planting area, and manure cost, as control variables (results for control variables were displayed in Appendix A Table A2).

Column (1) in Table 4 presents results for the OLS model, and showed that STFRP had a negative impact on TEFU. The counterintuitive sign may be a result of omitted variable bias. For columns (2) to (4), we added different levels of fixed effects to address this problem. Specifically, column (2) added the household-fixed effect, column (3) additionally included the year-fixed effect, and column (4) added the province-specific year-fixed effect. Column (4) was our preferred specification, since it not only accounted for household unobservables using household-fixed effects, but also accommodated temporal trends at the province level. The results of columns (2) to (4) showed that STFRP increased TEFU by about 0.006 (a roughly 4% increase from the average TEFU of 0.16 and about a 10% increase from the average TEFU of 0.06 in 2005). The results were fairly robust, especially when the household-fixed effect was included. We also estimated the impact of STFRP on wheat yield, and found that there was no impact of STFRP on yield; due to space limitations, the results of STFRP's impact on wheat yield were not shown in the paper, but were available upon request.

**Table 4.** Results of TEFU using a general DID approach.

| VARIABLES | (1) | (2) | (3) | (4) |
|---|---|---|---|---|
| T | −0.026 *** | 0.004 *** | 0.006 *** | 0.006 ** |
| | (0.002) | (0.002) | (0.002) | (0.002) |
| Constant | −0.058 | 0.261 *** | 0.257 *** | 0.596 |
| | (0.037) | (0.019) | (0.025) | (1.331) |
| Observations | 4249 | 4249 | 4249 | 4249 |
| R-squared | 0.273 | 0.655 | 0.660 | 0.665 |
| Number of households | | 1137 | 1137 | 1137 |
| **Controls** | | | | |
| Household and field characteristics | Yes | Yes | Yes | Yes |
| Household-fixed effect | | Yes | Yes | Yes |
| Year-fixed effect | | | Yes | Yes |
| Province-specific year-fixed effect | | | | Yes |

Note: Robust standard errors are clustered at the county level, and values are in parentheses: *** $p < 0.01$, ** $p < 0.05$. For more details on the impact of household and field characteristics on TEFU, please see Appendix A Table A2.

Considering that TEFU was constrained between zero and one, we also used a general DID-Tobit model to estimate the effect of STFRP (Appendix A Table A3). Column (4) is also our preferred specification. Columns (2)–(4) in Appendix A Table A3 show that STFRP increased TEFU by 2%–4%. The results in Appendix A Table A3 are very similar with those in Table 4, which proved that the STFRP was effective and that our results were stable across specifications.

### 3.3. Placebo Tests and Robustness Checks

Table 5 presents the results for two placebo tests and two additional robustness checks. Specifically, Model (1) is a timing falsification test that artificially sets implementation of STFRP to 2 years before the actual year of implementation. Model (2) is the placebo test using pesticide expenditure as a dependent variable, with the hypothesis that STFRP should not directly impact pesticide cost, since STFRP only focused on chemical fertilizers rather than pesticides. Column (3) uses the TEFU from the Cobb–Douglas production function as a dependent variable, and column (4) replicates our preferred specification, but only uses data from Henan Province, which is the largest wheat-producing province. Model (1) in Table 5 revealed that if we hypothetically moved the implementation time for STFRP two years earlier, there was no discernable impact on TEFU. Model (2) showed that STFRP had no impact on pesticide application. Model (3) showed that if we used the Cobb–Douglas production function to measure TEFU, the estimated policy effect was qualitatively the same. Model (4) showed that the effect of STFRP was about three times larger in the leading wheat-producing province than in others.

**Table 5.** Results of placebo tests and additional robustness checks.

| VARIABLES | 2 years later | Pesticide | TEFU-CD | Henan |
|---|---|---|---|---|
| T | −0.003 | −0.060 | 0.007 *** | 0.020 ** |
| | (0.003) | (0.078) | (0.002) | −0.008 |
| Constant | −2.231 *** | −51.245 | 12.969 *** | 2.916 |
| | (0.621) | (48.682) | (1.875) | −5.759 |
| Observations | 4249 | 4249 | 4249 | 1463 |
| R-squared | 0.662 | 0.209 | 0.701 | 0.676 |
| Number of households | 1137 | 1137 | 1137 | 321 |
| **Controls** | | | | |
| Household and field characteristics | Yes | Yes | Yes | Yes |
| Household-fixed effect | Yes | Yes | Yes | Yes |
| Province-specific year-fixed effect | Yes | Yes | Yes | Yes |

Note: TEFU-CD means TEFU is estimated using the Cobb–Douglas production function. Robust standard errors are clustered at the county level, and values are in parentheses: *** $p < 0.01$, ** $p < 0.05$.

## 4. Discussion and Conclusions

Using data from a nationally representative household survey and focusing on wheat production, this paper is the first to study the impact of China's nationwide fertilizer education program, STFRP, on the TEFU, specifically, using the SFA method to estimate the TEFU for wheat production, and then using the DID method to evaluate the impact of STFRP on TEFU.

Estimates from SFA showed that on average, the TEFU for wheat production in China was only 0.17; this suggested that current fertilizer use in wheat production was much higher than the optimal level. This result was consistent with previous studies [3,5,35]; for example, Wu's research [3] showed that the TEFU for agricultural production (including wheat, rice, corn and other crops) in 2007 was only 0.3 using household survey data from five provinces (Jilin, Heilongjiang, Zhejiang, Anhui and Sichuan). The research from Ma et al. [5] also provided the evidences for the low level of TEFU.

Both our descriptive statistics and results from DID regressions showed that STFRP had been effective in increasing TEFU. In particular, STFRP increased TEFU by 4% on average for farmers enrolled in the program versus those who were not. These results were robust to falsification tests on program timing and pesticide use, as well as multiple specification tests. The results in this paper were consistent with previous studies showing that programs that provide production guidance to farmers, such as the best nutrient management practices (BMPs) of the United States and the ICT projects of Africa, were effective. For example, Asenso-Okyere and Mekonnen's research [18] found that ICT projects in Africa have changed farmers' behavior, such as engaging in soil improvement by using composted manure and practicing better environmental conservation. Aker [36] studied the development of ICT for agricultural extension in developing countries and found that mobile phones could improve access to and use of information about agricultural technologies, potentially improving farmers' learning. Another example to prove the importance of ICT on agriculture in Africa is the information provided by mobile phones reduced by about 10 to 16 percentage of agricultural price dispersion in Niger, with a larger impact for those market pairs with higher transport costs [37]. The report from Arab Information and Communication Technologies Organization [38] also showed that ICT could play a key role in achieving the sustainable agricultural development and food safety if being used for information, knowledge and experience share between farmer communities though the world. To put the results of our findings in perspective, Huang et al. [8] found that intensive fertilizer training in Shandong Province reduced fertilizer usage by about 22% without reducing corn yield. This paper provides additional evidence that supports previous evaluations of STFRP conducted on a smaller scale. For example, only focusing on Taihu Basin in China, Luo et al. [39] found that STFRP reduces chemical fertilizer usage by about 7% for rice production. Ge and Zhou's [40] research showed that STFRP could reduce nitrogen fertilizer usage by about 38 kg/ha for agricultural production in Jiangsu Province.

Similar programs aimed at increasing agricultural input efficiency by providing information are gaining popularity around the world, especially in developing countries. For example, Pakistan's National Food Security Policy focuses on technology adoption to raise farm productivity [41]. The ICT projects in agriculture are initiated in the Arab countries [38]. Lessons from STFRP will be valuable for other developing counties that are considering similar policies.

This work has several methodological improvements compared to previous literature on agricultural input efficiency, chemical fertilizer use, and the evaluation of information provision programs. To our best knowledge, this study is the first to rigorously evaluate the impact of the nationwide STFRP on TEFU in wheat production. The study uses panel data from several major wheat-producing provinces, which is a significant improvement over previous studies that use cross-sectional data in a limited geographical area. Furthermore, our novel combination of productivity analysis and program evaluation methods exploiting micro-level farmer survey data makes an important contribution to both fields.

Two areas are beyond the scope of this paper but should be explored in future research. First, there are multiple facets of STFRP recommendations, including quantity, composition, and time.

Future research should explore the relative importance of these different components in order to provide suggestions to further improve STFRP. Second, this paper only focuses on TEFU, which is the first-order impact of the STFRP. Future studies should comprehensively evaluate the impacts of STFRP on crop yields, soil quality, and water quality to paint a complete picture of the impacts of STFRP.

**Author Contributions:** The authors contribute equally in the project.

**Funding:** This project is supported by the National Social Science Fund of China (Grant Nos. 17AJY019). The authors also appreciate the support from the USDA National Institute of Food and Agriculture Hatch project 101,030 and the ISU Center for China-US Agricultural Economics and Policy.

**Acknowledgments:** The authors greatly appreciate for editing assistance by Nathan Cook and Barbara Nodin of an earlier draft, and acknowledge the feedback and comments from the participants at AAEA 2018 annual conference, the Sustainability and Development Conference at University of Michigan, and the 2018 North American meeting of the Chinese Economists Society.

**Conflicts of Interest:** The authors declare no conflicts of interest.

## Appendix A

**Table A1.** Timing of STFRP implementation for our sample.

| Province | County | Year of STFRP Implement | Province | County | Year of STFRP Implement |
|---|---|---|---|---|---|
| Hubei | Xishui | 2005 | Shanxi | Gaoping | 2005 |
| Hubei | Tianmen | 2005 | Shanxi | Linyi | 2006 |
| Hubei | Nanzhang | 2005 | Shanxi | Pingding | 2006 |
| Hubei | Hanchuan | 2006 | Shanxi | Dingxiang | 2007 |
| Hubei | Xinzhou | 2006 | Shanxi | Pingshun | 2009 |
| Hubei | Jinzhou | 2006 | Shanxi | Liulin | 2010 |
| Hubei | Changyang | 2007 | Shanxi | Ying | 2010 |
| Hubei | Daye | 2007 | Sichuan | Jianyang | 2005 |
| Hubei | Xianan | 2008 | Sichuan | Fushun | 2005 |
| Hubei | Xuanen | 2008 | Sichuan | Jiange | 2005 |
| Hubei | Xaidian | 2008 | Sichuan | Pengxi | 2006 |
| Hubei | Xiangyang | 2009 | Sichuan | Langzhong | 2006 |
| Henan | Tanghe | 2005 | Sichuan | Pengzhou | 2007 |
| Henan | Taiqian | 2007 | Sichuan | Gulin | 2007 |
| Henan | Jiyuan | 2007 | Sichuan | Youxian | 2008 |
| Henan | Gongyi | 2007 | Sichuan | Jiangan | 2008 |
| Henan | Xincai | 2007 | Sichuan | Meishan | 2009 |
| Henan | Lushi | 2008 | | | |
| Henan | Yancheng | 2008 | | | |

**Table A2.** The impact of STFRP on TEFU using a general DID approach.

| VARIABLES | (1) | (2) | (3) | (4) |
|---|---|---|---|---|
| T | −0.026 *** | 0.004 * | 0.006 ** | 0.006 ** |
| | (0.002) | (0.002) | (0.002) | (0.002) |
| **Household Characteristics** | | | | |
| Age | 0.065 *** | 0.000 | 0.001 | 0.001 |
| | (0.007) | (0.003) | (0.003) | (0.002) |
| Edu | 0.014 *** | −0.001 | −0.001 | −0.000 |
| | (0.003) | (0.002) | (0.002) | (0.002) |
| Fasset | −0.003 *** | −0.001 | −0.001 | −0.000 |
| | (0.001) | (0.001) | (0.001) | (0.001) |
| Tland | −0.006 ** | 0.006 ** | 0.006 ** | 0.007 ** |
| | (0.003) | (0.003) | (0.003) | (0.003) |
| Tincome | −0.009 *** | 0.000 | 0.001 | 0.001 |
| | (0.003) | (0.001) | (0.001) | (0.001) |

**Table A2.** *Cont.*

| VARIABLES | (1) | (2) | (3) | (4) |
|---|---|---|---|---|
| **Field Characteristics** | | | | |
| Seedct | 0.008 *** | −0.004 *** | −0.006 *** | −0.006 *** |
| | (0.002) | (0.001) | (0.001) | (0.001) |
| Pesct | 0.004 *** | −0.000 | −0.000 | −0.001 |
| | (0.002) | (0.002) | (0.002) | (0.002) |
| Labor | −0.043 *** | −0.049 *** | −0.049 *** | −0.050 *** |
| | (0.002) | (0.007) | (0.007) | (0.007) |
| Manure | −0.010 *** | −0.003** | −0.003 *** | −0.004 *** |
| | (0.001) | (0.001) | (0.001) | (0.001) |
| Machine | 0.019 *** | −0.013 *** | −0.013 *** | −0.012 *** |
| | (0.002) | (0.002) | (0.002) | (0.002) |
| Irrigct | −0.003 *** | −0.004 *** | −0.003 *** | −0.003 *** |
| | (0.001) | (0.001) | (0.001) | (0.001) |
| Plantingarea | 0.024 *** | −0.003 | −0.003 | −0.004 * |
| | (0.002) | (0.003) | (0.002) | (0.002) |
| Constant | −0.058 | 0.261 *** | 0.257 *** | 0.596 |
| | (0.037) | (0.019) | (0.025) | (1.331) |
| Observations | 4249 | 4249 | 4249 | 4249 |
| R-squared | 0.273 | 0.655 | 0.660 | 0.665 |
| Number of households | 1137 | 1137 | 1137 | 1137 |
| **Fixed effects** | | | | |
| Household-fixed effect | | Yes | Yes | Yes |
| Year-fixed effect | | | Yes | Yes |
| Province-specific year-fixed effect | | | | Yes |

Note: Household characteristics include age (Age), education (Edu), fixed asset for agricultural production (Fassect), total land area (Tland), and total income (Tincome). Field characteristics include seed cost (Seedct), pesticide cost (Pesct), labor input (Labor), manure cost (Manure), machinery cost (Machine), irrigation cost (Irrigct), and wheat planting area (Plantingarea). Robust standard errors are in parentheses. *** $p < 0.01$, ** $p < 0.05$, and * $p < 0.1$. All regressions in this table use the technical efficiency of chemical fertilizer use (TEFU) as the dependent variable, and all include the logarithmic form of household characteristics and field characteristics. Column (1) uses the pooled ordinary least squares model and does not include fixed effects, and columns (2)–(4) include different fixed effects.

**Table A3.** Results using a general DID-Tobit approach.

| VARIABLES | (1) | (2) | (3) | (4) |
|---|---|---|---|---|
| T | −0.026 *** | 0.003 *** | 0.005 *** | 0.005 *** |
| | (0.003) | (0.001) | (0.001) | (0.001) |
| Constant | −0.058 * | 0.249 *** | 0.246 *** | 0.217 *** |
| | (0.031) | (0.014) | (0.015) | (0.015) |
| Rho | | 0.960 *** | 0.961 *** | 0.958 *** |
| Observations | 4249 | 4249 | 4249 | 4249 |
| Number of households | 1137 | 1137 | 1137 | 1137 |
| **Controls** | | | | |
| Household and field characteristics | Yes | Yes | Yes | Yes |
| Year-fixed effect | | | Yes | Yes |
| Province-specific year-fixed effect | | | | Yes |

Note: Robust standard errors are in parentheses. *** $p < 0.01$ and * $p < 0.1$.

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
