# Peer review of "Does Fertilizer Education Program Increase the Technical Efficiency of Chemical Fertilizer Use? Evidence from Wheat Production in China"

_sustainability, doi:10.3390/su11020543_

Reviewer 1 Report

Please write all results in the past tense (throughout the manuscript).

L58-61 - which fertilizer, please provide N-P-K content if any?

L90-91 - delete.

'joint stage' should be jointing stage.

Fig. 1, what is the x-axis? Please give axis information with units (if any).

Discussion/conclusion section should further be improved. L325-331 is the justification of the study, not the conclusion.  

Author Response

We have changed all the present tense into past tense when talking about results, throughout the paper.

Thanks for your suggestion. We have added the N-P-K content in line 60, deleted line 90 and line 91, changed the joint stage into jointing stage and explained what is the x-axis in a note for Figure 1.

For the conclusion, we have rewritten it again according to the style of the journal and added more appropriate references to compare our study with previous studies.

Reviewer 2 Report

Authors had improved the manuscript; However the discussion needs to the rewritten with more appropriate references. 

The reference format is not respected in the revised document. This need to be corrected.

Author Response

For the conclusion, we have rewritten it according to the Sustainability style and added more appropriate references to compare our study with previous studies. In addition, we also changed the format of reference according to the Sustainability’s guidance.

This manuscript is a resubmission of an earlier submission. The following is a list of the peer review reports and author responses from that submission.

Round  1

Reviewer 1 Report

Does fertilizer education program increase fertilizer use efficiency? Evidence from wheat production in China by Wang and Han.

This manuscript evaluates the impact of a nationwide project in China named STFRP, which was stated in 2005. The manuscript presents several flaws and as such, cannot be considered noble in science. In summary:

1) The manuscript reads like a project report rather than a research report. Authors, based on household survey, reported that the project increased fertilizer use efficiency (FUE) by 4% during the first 3 years. As such, this does not contribute anything to the science. First, FUE is a critically important concept in the evaluation of crop production systems and cannot be measured using a household survey approach. Second, as a reviewer, I want to know - why and how FUE was increased in project areas? Did the project implement any new approach on fertilizer use (quantity, quality, timing, method, etc)?, How was it measured?, What was the impact on soil quality and crop yields? But, I did not find my answers.

2)  The authors presented data collected during 2005 – 2008 to compare the differences between STFRP and non-STFRP areas, which have no value in the year of 2018. The project should have very different impacts at this point. Hence, this manuscript ended up as a simple “preliminary project report”.

3)   Another serious flaw is - the manuscript has no discussions. Authors should discuss their results based on the previous similar studies on FUE.

4)   The introduction is not focused on the topic and contains methods and results (P2L50-54), which is not valid.

5)   The authors used SFA model, but I did not see any approach – how the model was validated?

6)   More than half of the text under concussion section is not the conclusion; this is just a repetition of an introduction.

7)   English/grammar is too loose and needs significant revisions.

8)   The manuscript does not follow the heading order. For example, the first heading is “introduction”, the second heading is “background”, and the third heading is “empirical framework”…….etc. Please check the author’s guideline and format the manuscript accordingly.

9)   The use of Chinese local units for land area, income, etc. may not be valid internationally. 

Reviewer 2 Report

The subject is of great importance however, the research design need improvement. The results are not well presented and I wonder why authors present results on corn while dealing with wheat. The discussion needs to be improved.

The paper needs to be edited by native English speaking person.

Specific comments are on the marked copy of the manuscript. 
